MIT-CTP/5696

# Topological Dipole Insulator

Ho Tat Lam,[1, *] Jung Hoon Han,[2, †] and Yizhi You[3, ‡]

[1]*Department of Physics, Massachusetts Institute of Technology, Cambridge, Massachusetts 02139, USA*
[2]*Department of Physics, Sungkyunkwan University, Suwon 16419, South Korea*
[3]*Department of Physics, Northeastern University, 360 Huntington Ave, Boston, MA 02115, USA*
(Dated: March 22, 2024)

We expand the concept of two-dimensional topological insulators to encompass a novel category known as topological dipole insulators (TDIs), characterized by conserved dipole moments along the $x$-direction in addition to charge conservation. By generalizing Laughlin's flux insertion argument, we prove a no-go theorem and predict possible edge patterns and anomalies in a TDI with both charge $U^e(1)$ and dipole $U^d(1)$ symmetries. The edge of a TDI is characterized as a quadrupolar channel that displays a dipole $U^d(1)$ anomaly. A quantized amount of dipole gets transferred between the edges under the dipolar flux insertion, manifesting as 'quantized quadrupolar Hall effect' in TDIs. A microscopic coupled-wire Hamiltonian realizing the TDI is constructed by introducing a mutually commuting pair-hopping terms between wires to gap out all the bulk modes while preserving the dipole moment. The effective action at the quadrupolar edge can be derived from the wire model, with the corresponding bulk dipolar Chern-Simons response theory delineating the topological electromagnetic response in TDIs. Finally, we enrich our exploration of topological dipole insulators to the spinful case and construct a dipolar version of the quantum spin Hall effect, whose boundary evidences a mixed anomaly between spin and dipole symmetry. Effective bulk and the edge action for the dipolar quantum spin Hall insulator are constructed as well.

## CONTENTS

## I. INTRODUCTION

Topological insulators and quantum Hall states in two dimensions possess incompressible bulk with their boundaries hosting exotic gapless modes of chiral/helical charge currents [1–4]. These gapless edge states exhibit intriguing $U(1)$ quantum anomalies, precluding their realization in purely one-dimensional lattice models with local interactions. Over recent decades, considerable research has been dedicated to the quest for a more expansive range of topological insulators known as symmetry-protected topological phases (SPT) [5–35]. As demonstrated in [27, 36–39], the peculiar feature of topological insulators (or broadly defined SPTs) is the emergence of anomalous symmetry as the effective theory on the boundary. For instance, the quantum spin Hall insulator [4] whose boundary supports left- and right-moving charge channels with opposite $S_z$ spins manifests mixed quantum anomaly between the charge and spin $U(1)$ symmetries, as gauging the charge $U(1)$ symmetry inevitably leads to the breaking of spin $S_z$ conservation at the edge [27, 40–42].

More recently, an increasing amount of activity has been dedicated to the study of quantum many-body systems with conservation of higher multipolar moments in addition to the total charge [43–72]. Systems with multipole conservation law characteristically exhibit constrained dynamics for charged excitations [73–76] as they would inevitably violate the multipole symmetry constraint. Multipole conservation is also closely linked to glassy dynamics as it provides ways to achieve robust ergodicity breaking and anomalously slow diffusion [77–86], and plays a pertinent role in understanding experiments where ultracold atoms are prepared in strongly tilted optical lattices [87–89].

In this work, we take a further step in enlarging the scope of multipole symmetry consideration to topological quantum materials. We achieve this by constructing coupled-wire models for the dipolar version of the quantum Hall effect (denoted as *topological dipole insulator*, or TDI) for spinless fermions and quantum spin Hall insulators for spinful fermions. These coupled-wire models possess conserved charge and conserved dipole moment along one direction, are incompressible in the bulk, and host localized, gapless modes with quantum anoma-

---
* Electronic address: htlam@mit.edu
† Electronic address: hanjemme@gmail.com
‡ Electronic address: y.you@northeastern.edu

lies at their boundaries [42, 48, 64, 90–96]. The distinctiveness of TDI arises from the dipole conservation, which stringently forbids single-particle hopping in some specific directions. Both the spinless TDI and the spinful dipolar quantum spin Hall insulator can arise from multi-channel inter-wire scattering that preserves the dipole moment. By a careful choice of dipole symmetry-allowed inter-wire coupling terms, one can gap out all the modes in the bulk and leave a single *quadrupolar* channel gapless at each edge. The construction of a dipolar quantum spin Hall insulator proceeds in a similar manner, with extra considerations for the spin degrees of freedom at the wire.

Other pioneering works on the dipolar quantum Hall effect such as those proposed in [64, 97] assumed that dipole moments in both directions are conserved. In our model, dipole conservation occurs in only one direction, perpendicular to the edge in the case of a cylindrical geometry, which facilitates the microscopic wire construction. Thus our system possesses two $U(1)$ symmetries, $U^e(1)$ and $U^d(1)$, associated with the conservation of charge and the $x$-dipole moment, respectively.

We start by developing a generalization of Laughlin's flux-threading argument [98] into a no-go theorem in Sec. II, constraining the possible edge patterns in TDI and proposing a generalized Lieb-Schultz-Mattis theorem [99] to characterize potential quantum anomalies at the boundaries of TDI. In particular, we show that the $U^e(1)$ symmetry cannot be anomalous at the edge nor has a mixed anomaly with $U^d(1)$. As a result, in contrast with the integer quantum Hall effect which hosts one right-moving ($R$) channel at one edge and one left-moving ($L$) channel at the opposite edge, the edge of TDI is *forced* to host a quadrupolar edge pattern whose microscopic feature consists of one $R$-channel running along the $y$-direction at the coordinate $x = 1$, two $L$-channels (of different flavors) at $x = 2$, and another $R$-channel at $x = 3$. The opposite edge hosts its counter-propagate partner, with one $L$ channel, two $R$ channels, and another $L$ channel at $x = L_x - 2, L_x - 1$, and $L_x$, respectively, for a strip of width $L_x$. Such quadrupolar edge structures are in accord with the no-go theorem stating that, at the edge, a chiral channel with excitations carrying $U^d(1)$ charge is allowed but those with $U^e(1)$ charge are not.

The conclusion drawn from the general no-go argument is further supported by concrete microscopic construction for TDI in Sec. III A, where we generalize the coupled-wire scheme of Luttinger liquids [100–109] to incorporate dipole conservation. Note that in related works [93, 94, 96, 109, 110], various methods are demonstrated for constructing a broad class of 3D subsystem-symmetric topological phases, exhibiting fracton behavior, from a collection of one-dimensional subsystems, such as electronic quantum wires or spin chains. In Sec. III B, we analyze the effective theory of boundary excitations. As expected, the edge along the $y$-direction supports a quadrupole channel. By introducing interactions within the channel, we can gap out some edge excitations and leave behind a chiral mode that is charge-neutral but carries a dipole moment, and an anti-propagating chiral mode that is both charge and dipole-neutral. In App. A, we show that the same pattern of excitations appears at the edges

along the $x$-direction. For both the $x$- and $y$-edges, the chiral mode that carries the dipole moment is responsible for the $U^d(1)$ anomaly on the edge – A perturbative anomaly that triggers the increase/decrease of the dipole moment at the left/right edge under the $U^d(1)$ dipole flux insertion, resulting in the quantized transfer of the dipole across the bulk.

In Sec. III C, we derive dipolar Chern-Simons(CS) theory, effectively capturing the topological response of TDI. It turns out that the dipolar CS theory is reminiscent of the ordinary CS theory in the quantum Hall effect, but written in terms of gauge fields that transform according to the dipole symmetry of the underlying microscopic model [64]. The quantized coefficient of the dipolar CS theory naturally embodies the quantized dipole transport under the dipolar flux insertion. Finally, in Sec. IV, we extend our exploration of TDI to spinful fermions and construct a model for dipole quantum spin Hall insulator, whose boundary supports a mixed anomaly between dipole charge and the spin $S_z$ moment. The response theory is captured by a mutual Chern-Simons theory of the dipole and spin gauge field.

## II. FLUX INSERTION, EDGE PATTERNS, AND NO-GO THEOREM

Our objective is to explore the realm of two-dimensional (2D) topological insulators possessing the charge and dipole conservation,

$$Q = \int \rho(x,y) \, dxdy, \quad D_x = \int x\rho(x,y) \, dxdy \quad (2.1)$$

where $\rho(x,y)$ is the charge density. (The integral can be replaced by a sum over coordinates for lattice models.) A microscopic model satisfying these conservation laws would, for instance, consist of single-particle hopping along the $y$-direction but only dipole-conserving two-particle hopping along the $x$-direction,

$$t_c \sum_r \psi_r^\dagger \psi_{r+e_y} + t_d \sum_r \psi_{r-e_x} \psi_r^\dagger \psi_{r+e_x}^\dagger \psi_{r+2e_x} + h.c.$$

where $r = (x, y)$ and $e_i$ is the unit vector along the $i$-direction.

Here we are interested in a *gapped* and *topological* model with protected edge states, and the question of foremost importance is regarding the structure of gapless boundary modes in TDI, as well as the edge and bulk effective theories governing its low-energy response. These questions have been addressed to some extent in earlier formulations of quantum Hall states with dipole symmetry [64, 90] through field-theoretic methods, but here our approach differs significantly in that the dipole symmetry is explicitly imposed in only one direction. This scheme allows us to make a strong statement regarding the possible structure of edge states in TDI, construct the microscopic coupled-wire model, and derive the edge and the bulk effective theories rigorously from the coupled-wire model.

Given that the nontrivial topology of TDI can manifest itself through the symmetry anomaly on its boundary, we first

identify possible edge patterns that display quantum anomalies under the charge and dipole symmetries. Consider placing the TDI on a cylinder with open boundaries at $x = 1$ and $x = L_x$, in a lattice model where the $x$-coordinate takes integer values. Drawing from analogy to the integer quantum Hall state whose boundaries carry a left-moving chiral charge current at $x = 1$ and a right-moving one at $x = L_x$ respectively, a naive expectation for TDI is to simply 'double' the edge channels by placing 'chiral dipole patterns' on each edge, e.g. $R$-channel at $x = 1$, $L$-channel at $x = 2$; $L$-channel at $x = L_x - 1$ and $R$-channel at $x = L_x$. However, such arrangement of edge patterns violates dipole conservation and is forbidden, as can be demonstrated in a thought experiment similar to Laughlin's flux insertion argument for the integer quantum Hall effect [98], which we will elaborate on soon.

Since the TDI exhibits two $U(1)$ symmetries associated with the conservation of charge and $x$-dipole moment, two separate flux insertion processes denoted $U^e(1)$ and $U^d(1)$ can be envisioned corresponding to the change of the vector potential $A_y$ from zero to the following:

$$U^e(1): \ A_y = \frac{2\pi}{L_y}, \qquad U^d(1): \ A_y = \frac{2\pi x}{L_y}. \qquad (2.2)$$

The gauge potential $A_y$ minimally couples with the electron current through $e^{iA_y}\psi_r^\dagger \psi_{r+e_y}$ in the microscopic Hamiltonian, and $L_y$ is the circumference of the cylinder wrapped in the $y$-direction. The amount of flux inserted through the $x$-th ring on the cylinder is $\int_0^{L_y} A_y(x)\,dy$, equal to $2\pi$ (i.e., a flux quantum) for the $U^e(1)$ flux and $2\pi x$ for the $U^d(1)$ flux, respectively. Hence, the $U^e(1)$ flux insertion induces a charge shift of $\pm 1$ for the $R$ and $L$-modes, akin to the chiral anomaly in 1D Weyl fermions. The dipolar flux insertion $U^d(1)$ results in a charge shift of $\pm x$ for the $R$- and $L$-channel at the $x$-th wire.

For the dipolar edge patterns we conjectured earlier, the $U^d(1)$ flux insertion results in a charge shift of $+1, -2, -(L_x - 1), +L_x$ for the wires located at $x = 1, 2, L_x - 1, L_x$, respectively, resulting in the unit charge transfer from one edge to the other. This conserves the total charge but not the dipole moment, which changed by $2L_x - 4$ during the dipole flux insertion. Such a dipole insulator would be anomalous *in the bulk* and cannot be realized in 2D lattice models under local interactions. This exemplifies a no-go theorem, ruling out the existence of chiral dipole patterns at the boundary of a TDI.

With the preceding discussion in mind, we can systematically search for possible edge patterns of TDI that are *anomaly-free* under the $U^e(1)$ and $U^d(1)$ symmetries. In essence, each boundary may host gapless modes that are anomalous under either the charge $U^e(1)$ or the dipole $U^d(1)$ symmetry, but the combination of the two edges along with the bulk as a complete 2D theory must be anomaly-free. This requires that the bulk charge and dipole moment remain invariant after the flux insertions.

For the sake of argument, we assume that proximate to the edge, the $x = i$-th row contains $m_i$ copies of the chiral mode, as illustrated in Fig. 1. Here, the integers $m_i > 0$ ($m_i < 0$)

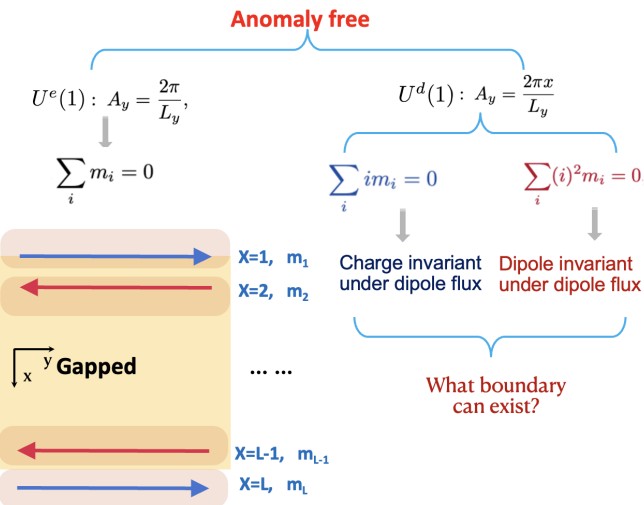

FIG. 1. Proximate to the edge, the $x = i$-th row contains $m_i$ copies of the chiral mode. The anomaly-free condition requires that the total charge and dipole moment remain invariant after $U^e(1)$ and $U^d(1)$ flux insertions, thus imposing constraints on the possible choices of $m_i$.

indicate an $R$-chiral ($L$-chiral) mode. Considering the localized nature of the edge modes, we postulate that nonzero $m_i$ is supported only on $1 \le i \lesssim \xi$ or $0 \le (L_x - i) \lesssim \xi$, where $\xi \ll L_x$ represents the correlation length.

The anomaly-free condition under the flux insertions can be formulated as algebraic relations among the integers $m_i$. The $U^e(1)$ flux insertion triggers a charge modification described by $(m_1, m_2, \ldots, m_{L_x-1}, m_{L_x})$ in the rows from $x = 1$ to $x = L_x$, as shown in Fig. 1. Since the theory requires the conservation of both charge and dipole moment, the total charge and dipole moment should remain invariant before and after the flux insertion:

$$\sum_{i=1}^{L_x} m_i = 0, \quad \sum_{i=1}^{L_x} i \times m_i = 0 \qquad (2.3)$$

In a similar vein, the dipole flux insertion via the gauge potential $A_y = \frac{2\pi x}{L_y}$ would lead to a charge modification described by $(m_1, 2m_2, 3m_3, \ldots, (L_x - 1)m_{L_x-1}, L_x m_{L_x})$. Consequently, the requirements of charge and dipole conservation after flux insertion impose two algebraic equations:

$$\sum_{i=1}^{L_x} i \times m_i = 0, \quad \sum_{i=1}^{L_x} i^2 \times m_i = 0 \qquad (2.4)$$

Overall the anomaly-free condition generates three sets of algebraic relations:

$$\sum_i m_i = \sum_i i \times m_i = \sum_i i^2 \times m_i = 0.$$

These relations constrain the possible edge patterns. Although numerous solutions exist, the simplest choice is:

$$(m_1, m_2, m_3) = (1, -2, 1)$$
$$(m_{L_x-2}, m_{L_x-1}, m_{L_x}) = (-1, 2, -1) \qquad (2.5)$$

Other solutions are either topologically equivalent to this one up to symmetry-allowed edge reconstructions or integer multiples of the above. The edge patterns implied by Eq. (2.5) can be viewed as a *quadrupole channel* at one edge and its time-reversal partner at the other. With these quadrupolar channels in place, both the charge and the dipole moment at each boundary remain unchanged under the $U^e(1)$ flux insertion. On the other hand, under the $U^d(1)$ flux insertion, the local dipole moment at the left/right edge increases/decreases by 2, resulting in the transfer of two units of dipole moment across the bulk under the dipolar flux insertion. This is the dipolar version of the Laughlin argument wherein a net charge transfer across the bulk takes place under the charge flux insertion. In other words, the edge quadrupole channels of TDI manifest a self-anomaly concerning the $U^d(1)$ symmetry.

Given that the TDI exhibits both charge and dipole conservations, one might wonder if a self-anomaly related to the $U^e(1)$ symmetry or a mixed anomaly between $U^d(1)$ and $U^e(1)$ might exist on the edge. We argue, contrary to intuition, that none of these can be realized in 2D lattice models. To see why, suppose there is a mixed anomaly between $U^d(1)$ and $U^e(1)$ symmetry at the edges. In this case, the flux insertion of $U^d(1)$ would alter the total charge at the left/right boundary by $m_L/m_R(=-m_L)$ respectively. However, such a charge transfer between the edges immediately violates the dipole moment conservation of the entire system, rendering the whole 2D theory anomalous. Therefore, a mixed anomaly between $U^d(1)$ and $U^e(1)$ cannot materialize at the edge of a 2D TDI. A similar reasoning rules out the $U^e(1)$ anomaly at the edge as well. In conclusion, the only conceivable anomaly at the edge of a dipole insulator is the perturbative anomaly related to the $U^d(1)$ symmetry. Similar restrictions on the boundary anomalies are also observed in 1D dipolar SPTs [67]. Our next step is to construct a microscopic model that reproduces such anomalous edge patterns of TDI.

### III. WIRE CONSTRUCTION OF TDI

We provide a microscopic model of TDI protected by charge and dipole conservations, by building upon the framework of coupled-wire construction [100, 102–105, 107, 109] which has proven vastly instrumental in developing various topological models with gapless boundaries. In particular, we adapt the coupled-wire construction for the quantum Hall states to incorporate the dipole conservation. The outcome is a coupled-wire model properly embodying the quadrupolar edge structure of TDI predicted in the previous section, and the edge and the bulk response theories for TDI.

#### A. Coupled-wire construction of TDI

As the minimal arrangement for constructing the TDI, we assume four flavors of right movers and another four flavors of left movers, for a total of eight wires per row $x \in \mathbb{Z}$. The low-energy kinetic Hamiltonian at $x$, with the velocity set to

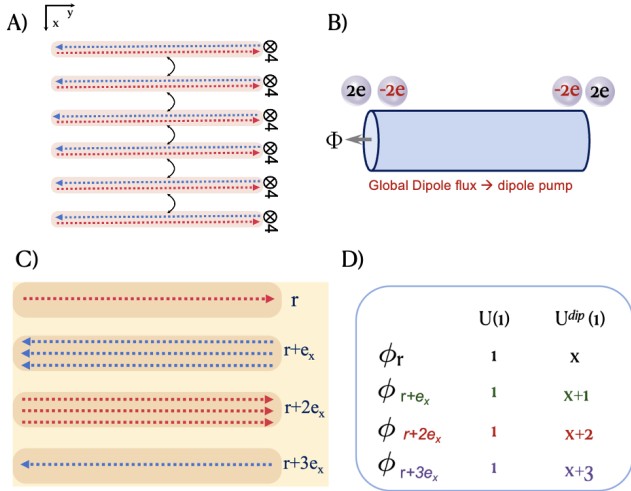

FIG. 2. A) Coupled wire setting for TDI: each row represents four flavors of 1D Dirac fermions of either $R$ (blue) or $L$ (red) chirality. B) Inserting a dipole flux through the cylinder triggers dipole pumping between the edges. C) The building block of the coupled-wire construction. D) How the charges in each wire transform under the $U^e(1)$ and $U^d(1)$ symmetries.

unity, is

$$\mathcal{H}_{\text{wires}}(x,y) = i \sum_{a=1}^{4} [\boldsymbol{\psi}^a(x,y)]^\dagger \tau^z \partial_y \boldsymbol{\psi}^a(x,y), \qquad (3.1)$$

spanning four flavors $a = 1, ..., 4$, and the Pauli matrix $\tau^z$ acts in the space of chirality corresponding to the left and right movers of 1D chiral fermions $\boldsymbol{\psi}^a = (\psi_L^a, \psi_R^a)$. All excitations are assumed to occur around the same lattice momentum.

Our objective is to figure out how to couple these 1D wires in a translation-invariant, dipole-preserving way so that all the bulk modes are gapped out. Recall the wire construction for the quantum Hall state: A pair of chiral modes $[\psi_L(x), \psi_R(x)]$ exists at each $x$. The inter-wire backscattering term $\sim \psi_L(x)\psi_R^\dagger(x+1) + h.c.$ is introduced between a pair of modes $[\psi_L(x), \psi_R(x+1)]$ and gaps out all the modes except the two chiral edge modes $\psi_R(1)$ and $\psi_L(L_x)$ as they are not involved in any of the backscattering terms.

To streamline this formulation, we select a set of chiral modes from several adjacent rows into a building block, e.g.

$$[\psi_L^1(x), \psi_R^{2,3,4}(x+1), \psi_L^{2,3,4}(x+2), \psi_R^1(x+3)]. \quad (3.2)$$

There is a degree of arbitrariness in the choice of flavor indices made above, which will not affect the final result. The full quantum wire array consists of these building blocks for $x \in \mathbb{Z}$, so that each mode $\psi_{L/R}^a$ only belongs to one unique building block. For TDI of width $L_x$, the building blocks are confined to $1 \le x \le L_x - 3$. The inter-wire coupling acts within each building block, and their effects can be analyzed independently of other building blocks.

As depicted in Fig. 2(C), we refer to $\psi_{L/R}^a(x)$ as the left/right $(L/R)$ chiral fermion mode of flavor $a$ at row $x$.

Henceforth, we will focus on selecting suitable inter-wire couplings within this building block to gap out the bulk. Single-particle hopping terms like $\psi_{L/R}^{\dagger,a'}(x')\psi_{L/R}^a(x)$ for $x' \neq x$ are prohibited due to dipole conservation. Consequently, the lowest-order inter-wire coupling terms are quartic in the fermion fields. The main challenge is to find a $U^d(1)$ symmetry-preserving coupling term that effectively gaps out all degrees of freedom within each block.

Hereafter, we will express the inter-wire coupling terms in the bosonized language, in which the fermion operator is expressed in terms of the vertex operator $\psi_{R/L}^a \sim e^{i\phi_{R/L}^a}$. The vertex operators transform under the $U^d(1)$ dipole symmetry as:

$$\phi_{L/R}^a(x,y) \to \phi_{L/R}^a(x,y) + \alpha \cdot x \qquad (3.3)$$

with arbitrary phase $\alpha$. The inter-wire coupling terms must be invariant under such transformations. Counting all the symmetry-preserving quartic tunneling terms gives the desired tunneling Hamiltonian,

$$
\begin{aligned}
\mathcal{V}(x) = \\
- g_1 &\cos[\phi_L^1(x) - \phi_R^2(x+1) - \phi_R^3(x+1) + \phi_L^4(x+2)] \\
- g_2 &\cos[\phi_L^1(x) - \phi_R^3(x+1) - \phi_R^4(x+1) + \phi_L^2(x+2)] \\
- g_3 &\cos[\phi_L^1(x) - \phi_R^4(x+1) - \phi_R^2(x+1) + \phi_L^3(x+2)] \\
- g_4 &\cos[\phi_R^2(x+1) - \phi_L^3(x+2) - \phi_L^4(x+2) + \phi_R^1(x+3)] \\
- g_5 &\cos[\phi_R^3(x+1) - \phi_L^4(x+2) - \phi_L^2(x+2) + \phi_R^1(x+3)] \\
- g_6 &\cos[\phi_R^4(x+1) - \phi_L^2(x+2) - \phi_L^3(x+2) + \phi_R^1(x+3)]
\end{aligned}
$$
$$(3.4)$$

We omit writing the $y$-coordinate labels explicitly in the Hamiltonian and assume that all cosine couplings in Eq. (3.4) occur at the same $y$. The coordinate $x$ ranges from $x = 1$ to $x = L_x - 3$, and we only include the chiral modes inside this building block. Recalling the commutation relations among the chiral fields:

$$
\begin{aligned}
[\phi_R^a(x,y), \phi_R^{a'}(x',y')] &= -[\phi_L^a(x,y), \phi_L^{a'}(x',y')] \\
&= \pi i \delta_{aa'}\delta_{xx'}\text{sgn}(y - y'), \quad (3.5)
\end{aligned}
$$

it is easy to check that all the cosine terms in Eq. (3.4) commute with each other. While six cosine terms are identified in Eq. (3.4), only the first four are independent; the last two being linear combinations of the first four. The inter-wire coupling detailed in Eq. (3.4) results in four independent mass terms, sufficient to gap out the four helical modes per building block and consequently all the bulk degrees of freedom, provided that the $g_i$ couplings are sufficiently strong.

A few modes do not show up in the inter-wire Hamiltonian (3.4) and continue to be gapless. For example, on the leftmost boundary at $x = 1$ we have only the $\phi_L^1(x = 1)$ mode present in Eq. (3.4), leaving the other seven modes gapless. However, generic intra-wire coupling within the $x = 1$ block can occur without violating the dipole conservation, allowing all three remaining $L$-modes to pair with three of the $R$-modes and gap each other out, ultimately leaving only one $R$-mode

gapless. At $x = 2$, the modes $\phi_R^1$ and $\phi_L^{2,3,4}$, not being part of the backscattering Hamiltonian, are gapless. Yet, $\phi_R^1$ can be gapped out against one of the three $L$-modes through intra-wire coupling, resulting in two gapless $L$-modes at $x = 2$. At $x = 3$, the only mode not included in the inter-wire Hamiltonian is $\phi_R^1$.

In the end, the modes that withstand all generic intra-wire gapping-out processes at the boundary are one $R$-mode at $x = 1$, two $L$-modes at $x = 2$, and one $R$-mode at $x = 3$, precisely reproducing the quadrupole channels anticipated from the general consideration of the previous section. It will be shown next that some of these modes can be further gapped out using inter-wire couplings and couplings to auxiliary neutral chiral modes.

### B. Edge effective theory

The hydrodynamic theory for edge excitations in the TDI can be formulated within the coupled-wire scheme. As established already, the gapless quadrupolar edge mode consists of $\phi_R^1(1)$, $\phi_L^2(2)$, $\phi_L^3(2)$, and $\phi_R^1(3)$. The effective Lagrangian for these boson modes contains the following terms:

$$
\begin{aligned}
\mathcal{L} = &+ \frac{1}{4\pi}\left(\partial_y\phi_R^1(1)\partial_t\phi_R^1(1) + \partial_y\phi_R^1(3)\partial_t\phi_R^1(3)\right) \\
&- \frac{1}{4\pi}\left(\partial_y\phi_L^2(2)\partial_t\phi_L^2(2) + \partial_y\phi_L^3(2)\partial_t\phi_L^3(2)\right) \quad (3.6)
\end{aligned}
$$

where the sign of each term reflects their chirality. It turns out that the following potential term, which commutes with all terms in $\mathcal{V}(x)$, can be added while observing the dipole symmetry:

$$V = -v\cos\left(\phi_R^1(1) - \phi_L^2(2) - \phi_L^3(2) + \phi_R^1(3)\right). \quad (3.7)$$

For sufficiently large $v$, the fields are restricted to the minimum of the potential:

$$\phi_L^3(2) = \phi_R^1(1) - \phi_L^2(2) + \phi_R^1(3).$$

Substituting this relation back to the Lagrangian in Eq. (3.6) and using integration by parts, we arrive at the new Lagrangian

$$\mathcal{L} = \frac{1}{2\pi}\partial_y\Phi_1\partial_t\Phi_2 \qquad (3.8)$$

where

$$\Phi_1 \equiv \phi_L^2(2) - \phi_R^1(1), \quad \Phi_2 \equiv \phi_R^1(3) - \phi_L^2(2).$$

These fields $\Phi_1, \Phi_2$ are invariant under the $U^e(1)$ symmetry but transforms under the $U^d(1)$ symmetry as $\Phi_{1,2} \to \Phi_{1,2}+\alpha$, leaving the Lagrangian Eq. (3.8) invariant under it.

We can finalize the construction of the effective edge theory of TDI by adding potential terms to quadratic order:

$$
\begin{aligned}
\mathcal{L} = &\frac{1}{2\pi}\partial_y\Phi_1\partial_t\Phi_2 - \frac{K'}{2\pi}\partial_y\Phi_1\partial_y\Phi_2 \\
&- \frac{K}{4\pi}\left(\partial_y\Phi_1\right)^2 - \frac{K}{4\pi}\left(\partial_y\Phi_2\right)^2. \quad (3.9)
\end{aligned}
$$

The coefficients must satisfy $K' < K$ to guarantee the stability of the potential. The ensuing equations of motion are

$$-\partial_y \partial_t \Phi_2 + K \partial_y^2 \Phi_1 + K' \partial_y^2 \Phi_2 = 0,$$
$$-\partial_y \partial_t \Phi_1 + K \partial_y^2 \Phi_2 + K' \partial_y^2 \Phi_1 = 0, \qquad (3.10)$$

solved by identifying two gapless modes

$$\Phi_L = \frac{1}{2}(\Phi_1 + \Phi_2), \quad \Phi_R = \frac{1}{2}(\Phi_1 - \Phi_2) \qquad (3.11)$$

with their respective dispersions $\omega_{L/R} = (K' \pm K)k$. Since $K' < K$, we conclude that the two eigenmodes are counter-propagating, and $\Phi_L, \Phi_R$ can be interpreted as the left and right moving modes. Among them, the right mover $\Phi_R$ is both charge- and dipole-neutral, and represents a neutral chiral mode. The left mover $\Phi_L$, on the other hand, transforms under the dipole symmetry, Eq. (3.3), as $\Phi_L \to \Phi_L + \alpha$. Only the $\Phi_L$ mode is responsible for the dipole anomaly at the boundary.

Finally, we comment on the edge stability and possible edge reconstruction. Although there are two counter-propagating modes $\Phi_L$ and $\Phi_R$ at the edge, they remain gapless under dipole symmetry as the backscattering between the dipole-charged mode $\Phi_L$ and the dipole-neutral mode $\Phi_R$ is prohibited. One can also consider more exotic edge reconstruction by adding layers of a chiral state that is both charge and dipole neutral (e.g., chiral spin liquid with a chiral central charge $c = 1$), to gap out the chiral neutral $\Phi_R$ mode and leave the boundary with only the chiral dipole current $\Phi_L$. The effective Lagrangian for the chiral dipole current is given by

$$\mathcal{L} = \frac{1}{2\pi} \partial_y \Phi_L \partial_t \Phi_L - \frac{1}{2\pi}(K + K')(\partial_y \Phi_L)^2. \qquad (3.12)$$

A concurrent question on the agenda concerns the effective theory of the edge along the $x$-axis. In the context of coupled wire construction in quantum Hall states, it is observed that both types of boundaries—whether parallel or perpendicular to the wires—exhibit the same edge state at the infrared (IR) limit, regardless of the anisotropy of the original Hamiltonian. However, this characteristic does not extend to our dipole-conserving quantum Hall phases. Specifically, along the $y$-boundary, the dipole symmetry behaves as if multiple independent $U(1)$ symmetries are acting on different rows near the edge. In contrast, at the $x$-edge, the $U^d(1)$ symmetry, acting along the $y$-column, performs as a genuine dipole symmetry in one dimension (1D). For completeness, we have worked out the edge effective action for the edge along the $x$-axis in Appendix A.

### C. Bulk effective theory

The $U^e(1)$ charge and the $U^d(1)$ dipole symmetry of the fermions in TDI dictates that they couple to background gauge fields with gauge symmetry given by

$$(A_t, A_{xx}, A_y) \to (A_t + \partial_t \lambda, A_{xx} + \Delta_x^2 \lambda, A_y + \partial_y \lambda). \qquad (3.13)$$

In accordance with the coupled-wire construction we employ the discrete derivative $\Delta_x$ defined as

$$\Delta_x \lambda(x) = \lambda(x+1) - \lambda(x),$$
$$\Delta_x^2 \lambda(x) = \lambda(x+1) - 2\lambda(x) + \lambda(x-1). \qquad (3.14)$$

Meanwhile, the bosonic fields transform under the gauge symmetry as $\phi_{L,R}(x) \to \phi_{L,R}(x) + \lambda(x)$. The response of the TDI is captured by a classical action of the background gauge field $(A_t, A_{xx}, A_y)$.

To derive the response theory, we first couple the wires to the background gauge field Eq. (3.13) and then integrate out the compact boson fields. It suffices to do this for a single building block. For instance, the first term in the potential Eq. (3.4) becomes

$$\cos[\phi_L^1(x) - \phi_R^2(x+1) - \phi_R^3(x+1) + \phi_L^4(x+2) - A_{xx}(x+1)]$$

under the coupling to the background field. At low energy, the compact boson fields are pinned to the minimum of the potential, satisfying

$$\begin{aligned}
\phi_R^2(x+1) &= \phi_L^1(x) + \phi_R^1(x+3) \\
&\quad - \phi_L^2(x+2) - A_{xx}(x+1) - A_{xx}(x+2), \\
\phi_R^3(x+1) &= \phi_L^1(x) + \phi_R^1(x+3) \\
&\quad - \phi_L^3(x+2) - A_{xx}(x+1) - A_{xx}(x+2), \\
\phi_R^4(x+1) &= \phi_L^2(x+2) \\
&\quad + \phi_L^3(x+2) - \phi_R^1(x+3) + A_{xx}(x+2), \\
\phi_L^4(x+2) &= \phi_L^1(x) + 2\phi_R^1(x+3) - \phi_L^2(x+2) \\
&\quad - \phi_L^3(x+2) - A_{xx}(x+1) - 2A_{xx}(x+2).
\end{aligned} \qquad (3.15)$$

The equations hold modulo $2\pi$ and we omit the $y$ and $t$ coordinates, which should be the same in all arguments. The kinetic term of the chiral bosons when coupled to the background gauge field is given by

$$\begin{aligned}
\mathcal{L} = &-\frac{1}{4\pi} \partial_y \phi_L^1(x)[\partial_t \phi_L^1(x) - 2A_t(x)] \\
&+ \frac{1}{4\pi} \partial_y \phi_R^1(x+3)[\partial_t \phi_R^1(x+3) - 2A_t(x+3)] \\
&+ \frac{1}{4\pi} \sum_{a=2,3,4} \partial_y \phi_R^a(x+1)[\partial_t \phi_R^a(x+1) - 2A_t(x+1)] \\
&- \frac{1}{4\pi} \sum_{a=2,3,4} \partial_y \phi_L^a(x+2)[\partial_t \phi_L^a(x+2) - 2A_t(x+2)] \\
&+ \frac{1}{2\pi} E_{xx}(x+1)[\partial_y \phi_L^1(x) - A_y(x)] \\
&- \frac{1}{2\pi} E_{xx}(x+2)[\partial_y \phi_R^1(x+3) - A_y(x+3)], \qquad (3.16)
\end{aligned}$$

where $E_{xx}(x) = \partial_t A_{xx}(x) - \Delta_x^2 A_t(x)$. The coupling to $A_t$ in the first four terms is the standard coupling of chiral bosons to background gauge fields (see, for example, App. A1 of [95]). The factor of 2 in front of $A_t$ is included such that the background gauge transformation of the Lagrangian is independent of the chiral boson fields. Although the Lagrangian is

not invariant under the background gauge transformation, the gauge variations in the action cancel each other after summing over all the blocks when there are no boundaries. The last two terms are already gauge-invariant, and the reason for including them will soon be clear. Substituting the solution Eq. (3.15) to the kinetic term Eq. (3.16) leads to the following response Lagrangian,

$$\mathcal{L} = +\frac{1}{2\pi}A_{xx}(x)\partial_y[A_t(x+1) - A_t(x-1)]$$
$$-\frac{1}{2\pi}A_{xx}(x)\partial_t[A_y(x+1) - A_y(x-1)]$$
$$-\frac{1}{2\pi}\Delta_x^2 A_t(x)[A_y(x+1) - A_y(x-1)], \quad (3.17)$$

where the simplicity of the final expression is due to rearranging terms from different building blocks and using integration by parts in the $y$ and $t$ directions. If the last two terms in Eq. (3.16) were not included, the above manipulation would not have led to a classical action, but rather an action that depends on some chiral boson fields.

We can take the continuum limit of the response theory Eq. (3.17) as follows. First, we introduce a lattice spacing $a$ between the wires and do the following change of variables

$$A_{xx}(x) \to a^2 A_{xx}(x), \quad \Delta_x^2 A_t(x) \to a^2 \partial_x^2 A_t(x),$$
$$A_{t,y}(x \pm 1) \to A_{t,y}(x) \pm a\partial_x A_{t,y}(x). \quad (3.18)$$

Next, we expand the response theory to the leading order in $a$. This gives the continuum response Lagrangian:

$$\mathcal{L} = \frac{2a^2}{2\pi}[A_{xx}\partial_y(\partial_x A_t) - A_{xx}\partial_t(\partial_x A_y) + (\partial_x A_t)\partial_x(\partial_x A_y)].$$
$$(3.19)$$

The coefficient is proportional to $a^2$ instead of $a^3$ since a factor of $a$ is absorbed into the sum, turning the latter into an integral. This Lagrangian can be repackaged into a more enlightening, Chern-Simons term

$$\mathcal{L} = \frac{k}{4\pi}\epsilon^{\mu\nu\rho}\mathcal{A}_\mu\partial_\nu\mathcal{A}_\rho \quad (3.20)$$

with level $k = 2$, in terms of a *dipolar* vector gauge field

$$\mathcal{A}_\mu = (\mathcal{A}_t, \mathcal{A}_x, \mathcal{A}_y) = (a\partial_x A_t, aA_{xx}, a\partial_x A_y). \quad (3.21)$$

The new gauge field has the gauge symmetry $\mathcal{A}_\mu \to \mathcal{A}_\mu + \partial_\mu\Lambda$, where $\Lambda = a\partial_x\lambda$, in contrast to (3.13). The factor $a$ cannot be removed arbitrarily from the definition of gauge transformations, and it records the UV cut-off. We refer to the response Lagrangian in Eq. (3.19) as a *dipolar Chern-Simons term*. Since the chiral bosons transform as $\phi \to \phi + \lambda$, the minimal coupling to the background gauge field takes the form $(\partial_\mu\partial_x\phi - \mathcal{A}_\mu)$ in the continuum theory. One can also define the *dipolar* electromagnetic fields as

$$\mathcal{B} = \partial_x\mathcal{A}_y - \partial_y\mathcal{A}_x = a(\partial_x^2 A_y - \partial_y A_{xx}),$$
$$\mathcal{E}_x = \partial_x\mathcal{A}_t - \partial_t\mathcal{A}_x = a(\partial_x^2 A_t - \partial_t A_{xx}),$$
$$\mathcal{E}_y = \partial_y\mathcal{A}_t - \partial_t\mathcal{A}_y = a\partial_x(\partial_y A_t - \partial_t A_y), \quad (3.22)$$

and write the dipolar Chern-Simons action as $\mathcal{L} \sim \mathcal{A}_t\mathcal{B} + \mathcal{A}_x\mathcal{E}_y - \mathcal{A}_y\mathcal{E}_x$. Despite the dipole symmetry, the response theory is still captured by a Chern-Simons theory, with the main difference from the ordinary quantum Hall state lying in the gauge transformation properties of the fields $\mathcal{A}_\mu$ [64].

The dipolar Chern-Simons action represents the quantized response of charge and dipole currents to the background fields. First, we compute the charge current in the $y$ direction by the variation of the response Lagrangian Eq. (3.19) with respect to $A_y$:

$$J_y^e = \frac{\delta\mathcal{L}}{\delta A_y} = \frac{2a}{2\pi}\partial_x\mathcal{E}_x, \quad (3.23)$$

where the superscript $(e)$ signifies the charge current. Accordingly, generating a current in the $y$-direction requires that the potential $A_t$ be at least cubic in the $x$-coordinate, $A_t \propto x^3$. By contrast, a charge Hall current in the conventional Hall effect requires only the linear potential $A_t \propto x$. A dipole, on the other hand, can move under the quadratic potential $A_t \propto x^2$ and the quadrupole, only under the cubic potential. Thus, the equation for $J_y^e$ reflects the building blocks of TDI being the quadrupolar channels as discussed earlier. We dub this phenomenon *quadrupolar Hall effect* for the charge transport along the $y$-direction in the TDI.

Varying the response Lagrangian with respect to $A_{xx}$ gives the current $J_{xx}$,

$$J_{xx} = \frac{\delta\mathcal{L}}{\delta A_{xx}} = \frac{2a^2}{2\pi}\partial_x E_y = \frac{2a}{2\pi}\mathcal{E}_y, \quad (3.24)$$

where $E_y = \partial_y A_t - \partial_t A_y$ is the usual electric field in the $y$-direction, not to be confused with the dipolar electric field $\mathcal{E}_y$. Despite the appealingly simple relation, one must exercise caution in interpreting $J_{xx}$. The current conservation equation arising from $U(1)$ dipole symmetry takes the form,

$$\partial_t\rho^e - \partial_x(\partial_x J_{xx}) + \partial_y J_y^e = 0, \quad (3.25)$$

suggesting that the charge current in the $x$ direction is

$$J_x^e = -\partial_x J_{xx} = -\frac{2a^2}{2\pi}\partial_x^2 E_y. \quad (3.26)$$

The charge Hall current in the $x$ direction of TDI depends on the second $x$-derivative of the electric field, $E_y \propto x^2$. The dipole current $J_x^d$ can be similarly read off from the dipole current conservation equation derived from Eq. (3.25),

$$\partial_t(x\rho) + \partial_x(J_{xx} - x\partial_x J_{xx}) + \partial_y(xJ_y) = 0. \quad (3.27)$$

The dipole current is

$$J_x^d = J_{xx} - x\partial_x J_{xx} = \frac{2a^2}{2\pi}(\partial_x E_y - x\partial_x^2 E_y). \quad (3.28)$$

It can be generated by an electric field linear in $x$, i.e. $E_y \propto x$. This is consistent with the analysis in Sec. II that inserting a dipole flux in TDI pumps dipoles from one edge to the other.

To complete the picture, the charge density $\rho^e$ follows from the response Lagrangian Eq. (3.19) as

$$\rho^e = \frac{\delta \mathcal{L}}{\delta A_t} = \frac{2a^2}{2\pi} \partial_x [\partial_y A_{xx} - \partial_x^2 A_y]$$
$$= -\frac{2a^2}{2\pi} \partial_x \mathcal{B}. \tag{3.29}$$

The various relations between the charge and dipole currents and the background fields represent the dipolar generalization of the well-known relation $(\rho^e, J_x^e, J_y^e) \propto (B, E_y, -E_x)$ in the integer quantum Hall effect.

We now comment on the quantization of the dipolar Chern-Simons term. Because the gauge parameter $\lambda$ appears in the gauge transformation Eq. (3.13) with a second order derivative in $x$, it allows an identification linear in $x$ as follows [55, 56]

$$\lambda \sim \lambda + 2\pi\mathbb{Z} + \frac{2\pi x}{a}\mathbb{Z}. \tag{3.30}$$

The coefficient for the linear identification is fixed such that the gauge parameter is $2\pi$ periodic on every wire. As a result, the gauge parameter $\Lambda$ for the vector gauge field $\mathcal{A}_\mu$ obeys the identification, $\Lambda \sim \Lambda + 2\pi$, which is the same identification for a gauge parameter of an ordinary compact vector gauge field. As the level $k$ for the Chern-Simons term is quantized to be an integer (on spin manifolds), so does the level $k$ of the dipolar Chern-Simons term.

Although the $k = 2$ level emerges naturally in our wire construction of TDI, we do not claim that this is the minimal dipole charge allowed and believe that a unit dipole ($k = 1$) is possible in a more intricate wire construction. Also, given the well-known extension of the wire construction to the fractional quantum Hall state [101], extension of our wire construction to the fractional TDI is a theme worth pursuing in the future.

## IV. DIPOLAR QUANTUM SPIN HALL INSULATOR

In this section, we propose a model for dipolar quantum spin Hall insulator possessing spin $U^s(1)$ symmetry for $S_z$ conservation in addition to the conservation of charge and dipole. A central question is whether the analog of quantum spin Hall state can be manifested in the presence of an additional $U^d(1)$ symmetry. As previously elucidated for the topological dipole insulator, a bulk topological state can be characterized by its anomalous boundary. Accordingly, our initial effort is to identify potential quantum anomalies at the boundary.

We begin by exploring the possibility of achieving an edge pattern that exhibits a mixed anomaly between $U^e(1)$ and $U^s(1)$. This evokes the prominent feature of the quantum spin Hall effect, wherein a flux insertion of the spin ($S_z$) results in a change in the charge at each boundary. What distinguishes our scenario is the additional requirement for the system to maintain $U^d(1)$ symmetry owing to the dipole conservation.

Let us suppose that a mixed anomaly does exist between spin $U^s(1)$ and charge $U^e(1)$ at the edge. This edge anomaly

can be manifested by inserting a spin flux given by $A_y^s = \frac{2\pi}{L_y}$. Such a mixed anomaly between $U^e(1)$ and $U^s(1)$ necessitates that the charge at the left and the right boundary change following the insertion of $U^s(1)$ flux. This then implies a charge transfer between the edges, violating the dipole conservation. Hence, it immediately follows that a gapped insulator with a mixed anomaly between $U^e(1)$ and $U^s(1)$ on the edge is impossible.

To avoid having the mixed anomaly between charge and spin, we can assume a pair of counter-propagating channels near the edges at row $x$ with one of these channels being spinful and having the chirality $m_i$, and the other spin-neutral channel having the opposite chirality $-m_i$. Since the opposite chiralities come in pairs at each edge row, their response to $U^e(1)$ or $U^d(1)$ is zero for each site $x$.

For concreteness, we can select $m_1 = 1$ and $m_2 = -1$ for the left edge, and $m_{L_x-1} = -1$ and $m_{L_x} = 1$ for the right edge, as depicted in Fig. 3. In this configuration, the leftmost edge ($x = 1$) hosts an $L$-channel carrying both charge and spin, and an $R$-channel that carries only charge. The edge at $x = 2$ features an $L$-channel with charge and an $R$-channel with both charge and spin. Such edge channel exhibits a mixed anomaly between $U^d(1)$ and $U^s(1)$. Under the insertion of a spin flux $A_y^s = \frac{2\pi}{L_y}$, the edge dipole moment increases (decreases) by 1 at the left (right) edge, resulting in the pumping of the dipole moment between the edges. This is a *dipolar quantum spin Hall effect*, wherein a unit dipole is transferred between boundaries following the insertion of spin flux.

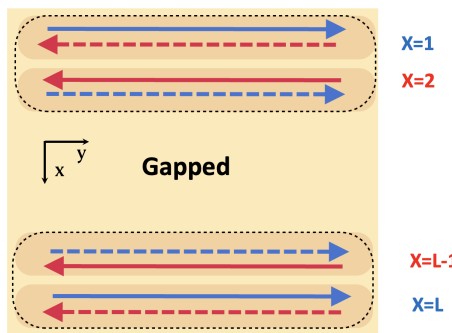

FIG. 3. Dipolar quantum spin Hall edge configuration. (The blue/red color indicates the L/R moving mode. The solid line refers to the mode that carries both charge and spin, while the dashed line only carries charge). At $x = 1$, there is a right-moving mode that carries both charge and spin (solid) accompanied by a left-moving mode (dashed line) that only carries charge. Similarly, the $x = 2$ row exhibits a right-moving mode that only carries charge, alongside a left-moving mode that carries both charge and spin.

A microscopic model supporting such anomalous edge states can be constructed using the coupled-wire construction. The 1D modes extended along the $y$-direction are $\phi_{L/R}^a(r), \Theta_{L/R}^a(r)$ ($a = 1, 2$ for flavor index), where the $\phi$-modes carry both charge and spin, and $\Theta$-modes carry only the charge. They transform under spin, charge, and dipole

symmetry as

$$U^e(1): \phi_{L/R}^a(x) \to \phi_{L/R}^a(x) + \gamma,$$
$$\Theta_{L/R}^{1,2}(x) \to \Theta_{L/R}^{1,2}(x) + \gamma,$$
$$U^s(1): \phi_{L/R}^a(x) \to \phi_{L/R}^a(x) + \beta,$$
$$\Theta_{L/R}^a(x) \to \Theta_{L/R}^a(x),$$
$$U^d(1): \phi_{L/R}^a(x) \to \phi_{L/R}^a(x) + \alpha \cdot x,$$
$$\Theta_{L/R}^a(x) \to \Theta_{L/R}^a(x) + \alpha \cdot x. \quad (4.1)$$

The elementary building block consists of the following eight modes (omitting $y$ coordinate):

$$\phi_L^1(x), \ \phi_R^{1,2}(x+1), \ \phi_L^2(x+2),$$
$$\Theta_R^1(x), \ \Theta_L^{1,2}(x+1), \ \Theta_R^2(x+2).$$

There are four spinless and four spinful modes in each block, extended over three adjacent $x$-coordinates and forming a quadrupolar channel. All the pertinent symmetries (charge, spin, dipole) are anomaly-free within the building block, so there are no obstructions to gapping them out while preserving the symmetries.

Four independent mass terms are required to gap out all the modes. In keeping with the dipole symmetry, we will couple the wires using quartic inter-wire interactions. After some consideration, we arrive at the tunneling Hamiltonian

$$\mathcal{V}(x) =$$
$$- g_1 \cos[\phi_L^1(x) - \phi_R^1(x+1) - \phi_R^2(x+1) + \phi_L^2(x+2)]$$
$$- g_2 \cos[\Theta_R^1(x) - \Theta_L^1(x+1) - \Theta_L^2(x+1) + \Theta_R^2(x+2)]$$
$$- g_3 \cos[\Theta_R^1(x) - \Theta_L^1(x+1) - \phi_L^1(x) + \phi_R^1(x+1)]$$
$$- g_4 \cos[\Theta_R^1(x) - \Theta_L^2(x+1) - \phi_L^1(x) + \phi_R^2(x+1)], \quad (4.2)$$

that respects all the symmetries defined in Eq. 4.1. All cosine terms in Eq. (4.2) are independent and commute with each other. At sufficiently strong coupling $g_i$, they are capable of generating four independent mass terms, leading to a fully gapped bulk.

At the boundary, there are eight chiral modes $\phi_R^{1,2}(1)$, $\phi_L^2(1)$, $\phi_R^1(2)$, $\Theta_L^{1,2}(1)$, $\Theta_R^2(1)$, $\Theta_L^1(2)$ that remain gapless. One can further introduce symmetry-permitted terms to couple these edge modes and gap out some degrees of freedom. First, we can add intra-wire coupling to gap out $\phi_R^2(1)$ against $\phi_L^2(1)$ and $\Theta_L^2(1)$ against $\Theta_R^2(1)$. This leaves four gapless chiral modes $\phi_L^1(1)$, $\phi_R^1(2)$, $\Theta_R^1(1)$, $\Theta_L^1(2)$ with the Lagrangian density

$$\mathcal{L} = - \frac{1}{4\pi} \partial_y \phi_L^1(1) \partial_t \phi_L^1(1) + \frac{1}{4\pi} \partial_y \phi_R^1(2) \partial_t \phi_R^1(2)$$
$$+ \frac{1}{4\pi} \partial_y \Theta_R^1(1) \partial_t \Theta_R^1(1) - \frac{1}{4\pi} \partial_y \Theta_L^1(2) \partial_t \Theta_L^1(2). \quad (4.3)$$

We can continue gapping out degrees of freedom by introducing the following inter-wire coupling:

$$V = -v \cos[\Theta_R^1(1) - \Theta_L^1(2) - \phi_L^1(1) + \phi_R^1(2)] \quad (4.4)$$

The hopping of the spinful mode from site 1 to 2 is compensated for by the hopping of the spinless mode from 2 to 1, thereby preserving the overall dipole moment. At strong enough $v$, the fields are pinned at $\Theta_R^1(1) = \Theta_L^1(2) + \phi_L^1(1) - \phi_R^1(2)$. Substituting this relation to the Lagrangian Eq. (4.3) leads to

$$\mathcal{L} = \frac{1}{2\pi} \left[ \partial_t \Phi \partial_y \tilde{\Phi} - \frac{K_1}{2} (\partial_y \tilde{\Phi})^2 - \frac{K_2}{2} (\partial_y \Phi)^2 \right], \quad (4.5)$$

where

$$\Phi = \phi_R^1(2) - \phi_L^1(1), \quad \tilde{\Phi} = \phi_R^1(2) - \Theta_L^1(2). \quad (4.6)$$

The last two terms in Eq. (4.5) are potential terms for $\Phi$ and $\tilde{\Phi}$, which we added by hand. This edge theory describes a Luttinger liquid, exhibiting an emergent 't Hooft anomaly, where the bosonic field $\Phi$ and its dual field $\tilde{\Phi}$ carry different symmetry charges. To be precise, based on the symmetry assignment in Eq. (4.1), $\Phi$ is spin- and charge-neutral and transforms under $U^d(1)$ as:

$$U^d(1): \Phi \to \Phi + \alpha \quad (4.7)$$

while $\tilde{\Phi}$ is dipole- and charge-neutral and transform under $U^s(1)$ as:

$$U^s(1): \tilde{\Phi} \to \tilde{\Phi} + \beta \quad (4.8)$$

This symmetry assignment in the edge theory reveals a mixed anomaly between dipole $U^d(1)$ and spins $U^s(1)$, which consequently prevents the gapping out of the helical mode in Eq. (4.5). By introducing a global $U^s(1)$ flux – realized through the addition of a spin gauge potential $A_y^s = \frac{2\pi}{L_y}$ – the system facilitates the transfer of a dipole moment from the left to the right boundary.

Equations of motion for $\Phi, \tilde{\Phi}$ are

$$\partial_t \partial_y \tilde{\Phi} - K_2 \partial_y^2 \Phi = 0,$$
$$\partial_t \partial_y \Phi - K_1 \partial_y^2 \tilde{\Phi} = 0. \quad (4.9)$$

There are two counter-propagating modes $\sqrt{K_2}\Phi \pm \sqrt{K_1}\tilde{\Phi}$ with the dispersions $\omega = \pm\sqrt{K_1 K_2}k$, respectively, representing the propagation of the mixed spin and dipole excitations. For $K_1 = K_2$, the two modes $\tilde{\Phi} + \Phi$ and $\tilde{\Phi} - \Phi$ share the same dipole moment but opposite spins.

The response of the dipolar quantum spin Hall effect is captured by a response functional of a pair of background gauge fields, one for the charge and dipole symmetry

$$(A_t, A_{xx}, A_y) \to (A_t + \partial_t \lambda, A_{xx} + \partial_x^2 \lambda, A_y + \partial_y \lambda) \quad (4.10)$$

and the other for the spin symmetry

$$(B_t, B_x, B_y) \to (B_t + \partial_t \chi, B_x + \partial_x \chi, B_y + \partial_y \chi). \quad (4.11)$$

Here, we directly work in the continuum limit so the gauge transformations use the continuum derivative $\partial_x$ instead of the

lattice difference $\Delta_x$. The response theory can be derived following a similar procedure as in Sec. III C, which gives

$$\mathcal{L} = \frac{a}{2\pi} B_t(\partial_x^2 A_y - \partial_y A_{xx}) + \frac{a}{2\pi} B_x \partial_x(\partial_y A_t - \partial_t A_y)$$
$$+ \frac{a}{2\pi} B_y(\partial_t A_{xx} - \partial_x^2 A_t), \qquad (4.12)$$

where $a$ is the lattice spacing between the wires. It can be repackaged into a mutual Chern-Simons theory of the two vector gauge fields $B_\mu = (B_t, B_x, B_y)$ and $\mathcal{A}_\mu = (a\partial_x A_t, a A_{xx}, a\partial_x A_y)$. It describes a *dipolar spin Hall effect*, where the spin Hall current in the $x$ direction,

$$J_x^s = \frac{\delta\mathcal{L}}{\delta B_x} = \frac{a}{2\pi}\partial_x E_y, \qquad (4.13)$$

is generated by a $y$ direction electric field $E_y = \partial_y A_t - \partial_t A_y$ that is linear in $x$, while the spin Hall current in the $y$ direction,

$$J_y^s = \frac{\delta\mathcal{L}}{\delta B_y} = \frac{a}{2\pi}(\partial_t A_{xx} - \partial_x^2 A_t), \qquad (4.14)$$

is generated by a potential quadratic in $x$.

Finally, we provide some insights into the mutual Chern-Simons response theory in Eq. 4.12. This theory intertwines a higher-rank gauge field, associated with the charge/dipole current, with a conventional vector gauge field associated with the spin current. The mutual Chern-Simons coupling suggests a mixed anomaly between the spin and dipole moments at the boundary. While the gauge potentials in this context are external fields probing electromagnetic responses, one might also consider a dynamical gauge field where the gauge currents represent the hydrodynamic charges, dipole, and spin currents of the quasiparticles. An intriguing question in this thrust is whether nontrivial braiding statistics can exist between spin excitations (which are fully mobile) and charge excitations (which have restricted mobility). Furthermore, it is worth exploring whether such braiding, between fully mobile excitations and fractons, could be captured through a mutual Chern-Simons coupling akin to Eq. 4.12. This line of investigation would enrich our understanding of the hybrid fracton phase [111, 112] from a field theory standpoint, and we reserve that for future explorations.

## V. SUMMARY AND OUTLOOK

We have proposed a notion of topological dipole insulators as an extension of quantum Hall and quantum spin Hall insulators by imposing a dipole symmetry in one direction. General anomaly considerations for the dipolar quantum Hall insulator show that it must host quadrupolar edge channels, with gapless excitations at the edge carrying quantized dipoles. A Luttinger-liquid wire model embodying the dipole symmetry is constructed employing charge- and dipole-conserving backscattering processes among the channels. Starting from the wire construction, one can construct the effective theory of edge dynamics of quantized dipoles, as well as the bulk response theory taking the form of the Chern-Simons action in

terms of dipolar gauge fields. In a model for dipolar quantum spin Hall insulator, an edge channel carrying one unit of dipole charge and one unit of spin $S_z$ exhibits mixed anomaly between the spin and dipole symmetries. A coupled-wire construction for such an insulator is also given, along with the edge effective action and the bulk response theory in the form of mutual Chern-Simons theory.

We speculate that a possible candidate for realizing this kind of topological dipole insulator could be the various two-dimensional moiré materials, which exhibit spontaneous quantum Hall ground states in the absence of magnetic field [113–118]. The moiré system's artificially large effective lattice spacing, on the order of the moiré length, and its shallow effective potential for the unit cell, make it significantly more susceptible to strong electric fields than ordinary atomic insulators. The electronic hopping under a strong tilted potential could lead to the emergence of a dipole conservation law, resulting in pairwise electron hopping, akin to what we envision in our wire construction. In a parallel thread, Ref. [119] proposed that lattice tilted by a strong linear potential and a weak quadratic potential naturally produce a rank-2 electric field, which is indeed coupled to the dipole current. Such tensor gauge fields can be realized in dipolar Harper-Hofstadter models in laboratories. We anticipate that the experimental setup for such a higher-rank electric field can enrich our explorations of the dipole pumping and topological responses in topological dipole insulators.

## ACKNOWLEDGMENTS

We are grateful to Gil-Young Cho, Debanjan Chowdhury, Meng Cheng, Cenke Xu for helpful discussions and feedback, and to Ethan Lake for collaborations on related work. J.H.H. is particularly grateful to Xiaoyang Huang for his insightful comments and discussion on dipolar Chern-Simons theory formulations. H.T.L. is supported in part by a Croucher fellowship from the Croucher Foundation, the Packard Foundation and the Center for Theoretical Physics at MIT. J.H.H. was supported by the National Research Foundation of Korea(NRF) grant funded by the Korea government(MSIT) (No. 2023R1A2C1002644). He also acknowledges financial support from EPIQS Moore theory centers at MIT and Harvard, where this work was initiated. This work was performed in part at Aspen Center for Physics (JHH, YY), which is supported by National Science Foundation grant PHY-2210452 and Durand Fund. This research was also supported in part by grants NSF PHY-1748958 and PHY-2309135 to the Kavli Institute for Theoretical Physics (KITP).

## Appendix A: Chiral quadrupole moment along the $x$-edge

In this appendix, we investigate the boundary theory of TDI by examining modes terminating in the direction perpendicular to the wires. In the context of coupled-wire construction across various quantum Hall states, both types of boundaries — whether parallel or perpendicular to the wires — exhibit

the same edge state dynamics in the infrared (IR) limit, regardless of the anisotropy of the original Hamiltonian. This characteristic does not extend to our dipole-conserving quantum Hall phases. Specifically, the dipole symmetry along the $y$-boundary behaves as if multiple independent $U(1)$ symmetries are acting on different rows near the edge. In contrast, for the $x$-edge, the $U^d(1)$ symmetry, acting along the $y$-column, functions as a genuine dipole symmetry in one dimension (1D). This raises a new question regarding the hydrodynamic description of the anomalous boundaries along the $x$-edge.

Since the wires terminate at the $x$-edge, we need to pick a boundary condition for the chiral bosons. A natural boundary condition that preserves the $U^e(1)$ and $U^d(1)$ symmetry is the reflected boundary condition that reflects a left-mover to a right-mover on the same wire. This can be realized by adding a boundary coupling between $\phi_L$ and $\phi_R$ as follows

$$
\begin{aligned}
S &= S_{\text{wire}} + S_{\text{boundary}}, \\
S_{\text{wire}} &= \frac{1}{4\pi} \int_{x<0} dx dt \, (\partial_y \phi_R \partial_t \phi_R - \partial_y \phi_L \partial_t \phi_L), \\
S_{\text{boundary}} &= \frac{1}{4\pi} \int_{x=0} dt \, \phi_L \partial_t \phi_R .
\end{aligned}
$$

Varying the action with respect to $\phi_L$, $\phi_R$ leads to the following boundary equations of motion

$$
\partial_t (\phi_L - \phi_R)|_{x=0} = 0, \tag{A1}
$$

that implements the reflected boundary condition dynamically. Since there are four pairs of left-mover and right-movers on each wire, we need four coupling terms on the boundary, which are chosen to be

$$
\mathcal{L} = \frac{1}{4\pi}(\phi_L^1 \partial_t \phi_R^4 + \phi_L^2 \partial_t \phi_R^3 + \phi_L^3 \partial_t \phi_R^2 + \phi_L^4 \partial_t \phi_R^1), \tag{A2}
$$

where all the fields live on the same wire sharing the same $x$ coordinate. Recall that there are inter-wire couplings in the bulk given by Eq. (3.4). At strong coupling $g_i$, the fields are

pinned down to the minimum of the potential,

$$
\begin{aligned}
\phi_R^2(x+1) &= \phi_L^1(x) + \phi_R^1(x+3) - \phi_L^2(x+2), \\
\phi_R^3(x+1) &= \phi_L^1(x) + \phi_R^1(x+3) - \phi_L^3(x+2), \\
\phi_R^4(x+1) &= \phi_L^2(x+2) + \phi_L^3(x+2) - \phi_R^1(x+3), \\
\phi_L^4(x+2) &= \phi_L^1(x) + 2\phi_R^1(x+3) - \phi_L^2(x+2) - \phi_L^3(x+2),
\end{aligned} \tag{A3}
$$

where the equations are valid modulo $2\pi$. Substituting this relation to the boundary Lagrangian Eq. (A2) and rearranging terms among different wires, we arrive at the following boundary Lagrangian

$$
\mathcal{L} = \frac{1}{4\pi} \partial_t \Phi_1(x) \left( \Phi_2(x+1) - \Phi_2(x-1) \right), \tag{A4}
$$

where we define

$$
\begin{aligned}
\Phi_1(x) &= \phi_R^1(x+1) - \phi_L^2(x), \\
\Phi_2(x) &= \phi_R^1(x+1) - \phi_L^3(x).
\end{aligned} \tag{A5}
$$

Both $\Phi_1$ and $\Phi_2$ are neutral under the charge symmetry $\phi(x) \to \phi(x) + \gamma$, but charged under the dipole symmetry $\phi(x) \to \phi(x) + \alpha x$ as

$$
\Phi_{1,2}(x) \to \Phi_{1,2}(x) + \alpha. \tag{A6}
$$

In the continuum limit, the boundary Lagrangian becomes,

$$
\begin{aligned}
\mathcal{L} = \frac{1}{2\pi} \partial_t \Phi_1 \partial_x \Phi_2 &- \frac{K'}{2\pi} \partial_y \Phi_1 \partial_y \Phi_2 \\
&- \frac{K}{4\pi} \left( \partial_x \Phi_1 \right)^2 - \frac{K}{4\pi} \left( \partial_x \Phi_2 \right)^2,
\end{aligned} \tag{A7}
$$

where we introduce a dipole-symmetry-preserving potential in the last three terms. The coefficients must satisfy $K' < K$ to guarantee the stability of the potential. The hydrodynamic equation governing the boundary modes are given by the equations of motion

$$
\begin{aligned}
\partial_t \partial_x \Phi_1 - K \partial_x^2 \Phi_2 - K' \partial_y^2 \Phi_1 &= 0, \\
\partial_t \partial_x \Phi_2 - K \partial_x^2 \Phi_1 - K' \partial_y^2 \Phi_2 &= 0.
\end{aligned} \tag{A8}
$$

There are two counter-propagating modes $\Phi_L = \frac{1}{2}(\Phi_1 + \Phi_2)$ with the dispersion $\omega = (K + K')k$ and $\Phi_R = \frac{1}{2}(\Phi_1 - \Phi_2)$ with the dispersion $\omega = -(K - K')k$. $\Phi_L$ carries one unit of charge under the $U^d(1)$ symmetry, representing a chiral dipole excitation, while $\Phi_R$ is neutral under all the pertinent symmetries and represents a neutral excitation with opposite chirality.

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
