# Peer review of "Topological Dipole Insulator"

_SciPost Physics_

## Round 1 · Referee Report · Anonymous (Referee 1) · 2024-6-15

Strengths

1- proposes new kinds of topological insulator and quantum spin Hall insulator protected by dipole symmetry in 2+1d.

2- constructs explicit models which realise these new (SPT) phases.

Report

This paper proposes a new kind of topological insulator, referred to as topological dipole insular, and quantum spin Hall insulator with conserved charge and dipole moment. The anomalies at the boundary and possible edge patterns are explored. Explicit microscopic models realising these edge patterns are given. The SPT action in the bulk corresponds to a higher-rank gauge theory which has a Chern-Simons or mutual Chern-Simons (BF) like description.

I believe the paper certainly meets the criteria for publication in SciPost Physics. I have a couple of minor comments listed below.

Requested changes

1- In the introduction, the authors say "... proposing a generalized Lieb-Schultz-Mattis theorem...," but I didn't see it being discussed in the rest of the paper. Can the authors please clarify what they meant to say?

2- In Fig 1 and 3, $L$ should be $L_x$.

3- In Fig 2, I think the x and y directions in (B) are rotated by 90 degrees with respect to (A) and (C).

Recommendation

Publish (easily meets expectations and criteria for this Journal; among top 50%)

  • validity: top
  • significance: high
  • originality: top
  • clarity: top
  • formatting: perfect
  • grammar: excellent

Author:  Ho Tat Lam  on 2024-10-09  [id 4848]

(in reply to Report 1 on 2024-06-15)

We thank the reviewer for valuable comments and suggestions.

1- In the introduction, the authors say "... proposing a generalized Lieb-Schultz-Mattis theorem...," but I didn't see it being discussed in the rest of the paper. Can the authors please clarify what they meant to say?

We apologize for the confusion. The generalized LSM theorem refers to our flux insertion argument, which identifies a no-go theorem and predicts possible edge patterns. These two concepts have been interrelated for historical reasons, but we agree that the term 'LSM theorem' might be confusing. In the revised version, we have replaced it with 'flux insertion argument'.

2- In Fig 1 and 3, $L$ should be $L_x$.

In the revised version, we have corrected the figure labeling for Fig. 1 and 3

3- In Fig 2, I think the x and y directions in (B) are rotated by 90 degrees with respect to (A) and (C).

In the revised version, we have unified the axis in Fig. 2.

---

## Round 1 · Referee Report · Anonymous (Referee 2) · 2024-9-6

Report

This paper introduces and explores the concept of topological dipole insulators (TDIs), a novel class of two-dimensional topological phases that conserve both charge and dipole moment along one direction. The authors provide a comprehensive theoretical study for these systems, from general arguments to microscopic models and effective field theories. They also extend the concept to spinful systems, introducing a dipolar version of the quantum spin Hall effect.

The paper's strength lies in its thorough theoretical treatment, which includes:

  1. General arguments based on flux insertion, generalizing Laughlin's approach to the dipole-conserving scenario.
  2. Microscopic coupled-wire constructions that explicitly realize the TDI phase.
  3. Effective field theories for both bulk and edge, including a "dipolar Chern-Simons" theory for the bulk response.

The presentation is well-structured, starting from general considerations and progressing to specific models and their properties. This logical flow helps the reader grasp the concept of TDIs and their unique features step by step.

One of the paper's key results is the prediction of "quadrupolar" edge channels in TDIs, which is derived from general anomaly considerations and then explicitly realized in the microscopic model. The extension to spinful systems, resulting in a dipolar quantum spin Hall effect, further demonstrates the richness of this new class of topological phases.

Minor question: In Eq. (2.5), the authors provide a solution for the anomaly-free condition and argue that this is the most general form. Can the authors elaborate on this point? What are the symmetry-allowed possible edge reconstructions? Could the authors provide an example or two?

Overall, this paper represents an intriguing result in the field of topological phases of matter with modulated symmetry and opens up new directions for future research. I am happy to recommend this paper for publication.

Recommendation

Publish (easily meets expectations and criteria for this Journal; among top 50%)

  • validity: top
  • significance: high
  • originality: high
  • clarity: top
  • formatting: perfect
  • grammar: perfect

Author:  Ho Tat Lam  on 2024-10-09  [id 4849]

(in reply to Report 2 on 2024-09-06)

We thank the reviewer for the valuable comments and questions.

In Sec III-B, we presented a concrete edge reconstruction that gaps out partial degrees of freedom at the boundary while preserving the key features of the anomalous boundary. This is a concrete example to show that while the microscopic pattern of the edge is different, the dipole-anomaly, and the gapless mode that is responsible for the dipole anomaly persist regardless of edge reconstruction. In addition, around equation 3.12, we discussed another exotic edge reconstruction by adding a layer of chiral state that is both charge and dipole neutral (e.g., chiral spin liquid with chiral central charge $c=1$) and gapping out $\Phi_R$ against this additional chiral mode. As the result, the edge effectively only consists of the chiral mode $\Phi _L$ which carry dipole current. In the revised version, we added two sentences at the end of the paragraph after eq 2.5 referring the readers to Sec III-B where we have extended discussions.

---

## Editorial Decision

resubmitted